# Sustainability Assessment of Energy Storage Technologies Based on Commercialization Viability: MCDM Model

Xiaoyang Shu [1], Raman Kumar [2,*], Rajeev Kumar Saha [3], Nikhil Dev [3], Željko Stević [4,*], Shubham Sharma [5,6] and Mohammad Rafighi [7]

1  School of Computer and Artificial Intelligence, Southwestern University of Finance and Economics, Wenjiang District, Chengdu 611130, China
2  Department of Mechanical and Production Engineering, Guru Nanak Dev Engineering College, Ludhiana 141006, Punjab, India
3  Department of Mechanical Engineering, J.C. Bose University of Science and Technology, YMCA, Faridabad 121006, Haryana, India
4  Faculty of Transport and Traffic Engineering, University of East Sarajevo, 74000 Doboj, Bosnia and Herzegovina
5  Mechanical Engineering Department, University Centre for Research and Development, Chandigarh University, Mohali 140413, Punjab, India
6  School of Mechanical and Automotive Engineering, Qingdao University of Technology, Qingdao 266520, China
7  Department of Aeronautical Engineering, Sivas University of Science and Technology, Sivas 58000, Türkiye
*  Correspondence: sehgal91@gndec.ac.in (R.K.); zeljkostevic88@yahoo.com or zeljko.stevic@sf.ues.rs.ba (Ž.S.)

**Abstract:** Advances in developed and developing countries are more attributable to growth in industrial activities that directly impact increasing energy demand. Energy availability has been inconsistent globally, necessitating energy storage (ES) for use as per requirement. Various energy storage technologies (ESTs) are available in mechanical, electrochemical, electrical, chemical, and thermal forms to fulfil the energy demand of a user as and when required. The factors responsible for making a commercially viable energy storage product are further being researched for an eco-friendly and optimal solution to store energy for a longer duration. Researchers are employing different strategies to evaluate the energy efficiency of storage technologies. This paper uses the VIKOR technique to analyze ESTs while assigning objective weights with the entropy weights method based on identified energy performance indicators and ranking them according to their commercialization viability. The method helps a consumer choose better ESTs as per their requirement while manufacturers compete with each other to enhance the commercial value of their energy storage products. Sensitivity analysis has been performed to understand the uncertainties, pros, and cons with the limitations and scope of using the decision model and thus taking an informed decision. The analysis of different energy storage technologies has indicated Hydrogen Fuel Cells (HFC) to be impressive and promising for the future.

**Keywords:** energy storage technologies; energy efficiency; energy performance indicators; entropy and VIKOR; Multi Criteria Decision Making (MCDM)

## 1. Introduction

Energy can be attributed as one of the primary driving forces for the economic development of a country. The industrial sector is dependent on a continuous supply of energy to work efficiently. Primary energy consumption worldwide has reached over 634 exajoules as of 2020 and is expected to increase to 935 exajoules by 2050, an average annual percentage change of around 1.3% [1]. The electricity generation in 2020 was 24,993 billion kilowatt-hours which are expected to be 41,953 billion kilowatt-hours by 2050, showing an average annual percentage change of around 1.7% [2].

Primary energy sources are gradually changing from non-renewable to renewable, resulting in smaller carbon footprints. However, the obtainability of continuous energy from renewable sources for consumers is a major concern. The problem that is being faced is the availability of energy as per demand. The energy demand can be fulfilled by storing it in bulk, using it whenever required, and restoring utilized energy in a shorter period. This will balance the energy demand and supply continuously [3]. The International Energy Agency (IEA) has presented a hypothetical scenario of Net Zero Emissions by 2050 (NZE). This hypothetical IEA scenario illustrates a constrained but attainable path for net zero emissions by 2050 [4].

The creation and use of renewable energy sources, such as solar and wind power, have significantly increased in recent years. While clean and plentiful, these sources are also sporadic and unpredictable, which can present problems for the stability and dependability of the electricity system [5]. By storing surplus energy generated during low demand and releasing it during high demand, energy storage devices can address these issues. Thus, several energy storage systems have been developed due to the rising global energy demand and the requirement to minimize carbon emissions. These technologies offer a way to store and release energy when needed, which is essential for incorporating renewable energy sources into the electrical grid [6]. With so many energy storage alternatives available, assessing and contrasting them based on their sustainability and financial viability is crucial [7].

Nevertheless, there is not a single EST that works for everyone. Each technology has advantages and disadvantages; some are better suited to particular conditions or uses. For instance, pumped hydro storage is suitable for large-scale energy storage, whereas lithium-ion batteries are suitable for handheld gadgets and electric cars. Several factors must be considered, including each technology's environmental impact, cost-effectiveness, and scalability, to identify the most sustainable and economically feasible ESTs [8].

Multi-Criteria Decision Making (MCDM) models can be used in this situation. MCDM is a powerful analytical technique that ranks and assesses several possibilities based on several distinct criteria or aspects. MCDM can evaluate and compare the viability and commercial viability of various energy storage systems [9]. Based on the chosen criteria, the MCDM model may evaluate and compare the available energy storage technologies, thoroughly evaluating their sustainability and commercial viability [10]. The MCDM model is an effective tool for evaluating these criteria and making informed decisions [11]. The MCDM model considers multiple criteria simultaneously, using mathematical algorithms to assign weights and scores to each criterion [12]. The model then ranks the energy storage technologies based on performance, allowing for an objective and comprehensive assessment [13]. Policymakers, investors, and energy firms can utilize the MCDM model's findings to guide their decisions on deploying ESTs in the future. This, in turn, can direct investment choices and encourage the use of ESTs, which can assist in overcoming the difficulties of integrating renewable energy sources and lowering greenhouse gas emissions. This study uses MCDM models to examine the sustainability of various energy storage systems. This paper has been written to find the most promising EST in terms of sustainable energy and eco-friendly for their commercial viability on a long-term basis.

## 2. Related Work: Review of Literature

Saaty introduced the Analytic Hierarchy Process (AHP), which is a widely used MCDM method for structuring decision problems into a hierarchy of criteria and alternatives, and then using pairwise comparisons to derive weights for each criterion and alternative [14]. Keeney & Raiffa [15] provided a comprehensive overview of MCDM methods and techniques and emphasized the importance of considering trade-offs and preferences when making decisions with multiple objectives. Roy [16] presented a systematic and comprehensive approach to MCDM, including an overview of different MCDM methods, their applications, and their limitations. Zavadskas et al. [17] provided an overview of different MCDM methods and their applications in economics [18,19] and included a discussion of

the strengths and weaknesses of each technique. MCDM is a complex and diverse field that requires careful consideration of multiple criteria and objectives. There are many different MCDM methods and techniques available, and the choice of method depends on the specific needs of the decision problem. VIKOR (VlseKriterijumska Optimizacija i Kompromisno Resenje) is a popular multi-criteria decision-making (MCDM) method for ranking alternatives based on multiple criteria.

Opricovic & Tzeng [20] introduced the VIKOR method and applied it to post-earthquake reconstruction planning. The authors demonstrate the effectiveness of the VIKOR method in balancing conflicting criteria and producing compromise solutions. Yang et al. [21] proposed a modified VIKOR method for supplier selection incorporating weights for the decision criteria. The authors demonstrate the effectiveness of the modified VIKOR method in selecting the best supplier among multiple alternatives. Shemshadi et al. [22] proposed a fuzzy VIKOR method for supplier selection incorporating entropy weight. The authors demonstrate the effectiveness of the fuzzy VIKOR method in selecting the best supplier among multiple alternatives. The VIKOR method was applied to the problem of employee selection. The authors demonstrate the effectiveness of the VIKOR method in selecting the best employee among multiple candidates [23].

Mardani et al. [24] reviewed the VIKOR method's methodology, applications, and performance comprehensively. The authors demonstrate the effectiveness of the VIKOR method in various fields, including engineering, management, and environmental sciences. Kahraman et al. [25] applied VIKOR to the problem of comparing catering service companies based on multiple criteria. The authors demonstrate that VIKOR is an effective method for ranking alternatives based on multiple criteria. Zhang et al. [26] proposed an extension of VIKOR that uses intuitionistic fuzzy sets to handle uncertainty in decision-making problems. The authors demonstrated the effectiveness of the proposed method through a numerical example.

Zhao et al. [27] suggested a united fuzzy-MCDM with 15 sub-criteria for thoroughly evaluating battery ES systems. Streimikiene et al. [28] established the MCDM methodology for choosing the utmost sustainable power generation technology, encompassing several sustainability perspectives. The study demonstrates that renewable electricity strategy should prioritize sustainable energy options such as water and solar thermal. Li et al. [29] evaluated electrochemical ES using MCDM for optimal program choosing out of five options and gave provisions related to stakeholders and decision-makers. Colak and Kaya [8] analyzed ESTs for Turkey by considering 9 alternatives, 4 attributes, and 19 sub-attributes. The economic and technical factors were decided to be the greatest and least essential major criteria, respectively, and the option of Compressed Air was judged to be the most suited. Albawab et al. [30] proposed a sustainability measurement model to rate ESTs. To assess the efficiency and efficacy of the suggested hybrid technique, 5 primary sustainability criteria and 17 sub-criteria were utilized, along with 7 ESTs. The results suggest that thermal ES and compressed air alternated between first and second place, and the MCDM approach is an effective tool for determining sustainability indicators for evaluating ESTs. lbahar et al. [31] used VIKOR and DEMATEL to evaluate the best hydrogen ES solution based on economic, technological, environmental, and social variables. The results suggested that storage capacity and response time were the most and least essential criteria, respectively, and that carbon nanotubes were the best option for Turkey. It is a useful pathway for academics and politicians interested in hydrogen storage. Kizielewicz et al. [32] introduced the MCDM method for selecting the most beneficial ES version from numerous accessible options. With the increased popularity of solar systems, there is a surge in interest in ES. Since of the numerous technical aspects of ES systems, the MCDM technique is preferred because it allows for quality rating while considering several factors, despite the absence of specialist knowledge of these equipment. The adoption of renewable storage technologies is critical for preserving sustainable electricity supply and demand balance, lowering the implementation cost of newly developed energy, and speeding the tempo of the new energy transition. In Jiangsu Province, China, Liu and Du [33] suggested

a MCDM architecture for renewable ES technology assortment from the standpoint of collective decision-making. It aided managers in making more scientific decisions on renewable EST alternatives.

One of the key criteria used in evaluating ESTs is energy density and other important criteria include power density, round-trip efficiency, life cycle cost, and environmental impact. Some studies have also considered criteria such as scalability, safety, and ease of maintenance. Several studies have used MCDM methods to evaluate and compare different ESTs. The results of these studies have varied, depending on the specific methods used, the criteria considered, and the weighting of the criteria.

There is a need to apply MCDM to analyze different ESTs as the complex and dynamic nature of the ES sector drives them. ESTs are diverse, and each technology has its own set of strengths and weaknesses, making it difficult to compare and evaluate them based on a single criterion, such as cost or efficiency. MCDM provides a systematic and rigorous approach to analyzing and comparing different ESTs based on multiple criteria, allowing decision-makers to make informed choices about which technology to adopt. MCDM considers multiple criteria such as cost, energy efficiency, safety, environmental impact, scalability, etc., and uses mathematical models to rank the technologies based on their performance against these criteria. This provides a comprehensive and objective evaluation of the different ESTs and helps identify the technology most suitable for a specific application or context. MCDM in ES analysis is essential in today's rapidly changing energy landscape, where the need for more sustainable and efficient ES solutions proliferates. MCDM provides a valuable option for decision-makers to assess the potential of different ES technologies and identify the best path forward for developing and deploying sustainable ES solutions. So, in the present work, the VIKOR method is applied to assign ranks to different ES technologies while considering 16 ESTs against 11 parameters and conducting sensitivity analysis.

## 3. Materials and Methods

### 3.1. Description of the Problem

The workflow chart of the present decision-making work to rank different ETS is shown in Figure 1. It consists of an exhaustive literature review of EST, and decision-making techniques applied in EST are discussed in Sections 1 and 2, respectively. After review, 16 ESTs were identified for decision-making, and details are described in Section 3.1.1 while considering 11 criteria described in Section 3.1.2. The VIKOR method is applied to rank ESTs (Section 3.2.1), the Entropy method to assign objective weights to 11 criteria (Section 3.2.2) and the Delhi method to attain expert opinion (Section 3.2.3). The implementation of decision-making to assign ranks to ESTs is presented in Section 4. Sensitivity analysis and implications of the study are presented in Section 5, and finally, conclusions are in Section 6.

#### 3.1.1. Energy Storage Technologies

Several energy storage technologies are available today, each with advantages and limitations. Various methods have been employed to store the energy long-term, though a plausible solution is yet to be successfully employed and commercially exploited. To get insight into promising ESTs, they have been classified and shown in Figure 2 [34,35]. Various ESTs are being researched for their commercial viability on a long-term basis. A few promising ES technologies are outlined in Figure 2 and are discussed here.

Pumped Hydro Energy Storage (PHES): PHES works by using excess electricity generated during periods of low demand to pump water from a lower reservoir to an upper reservoir. PHES remains an important technology for ES and is expected to play a noteworthy part in the upcoming transition to a more sustainable energy [36].

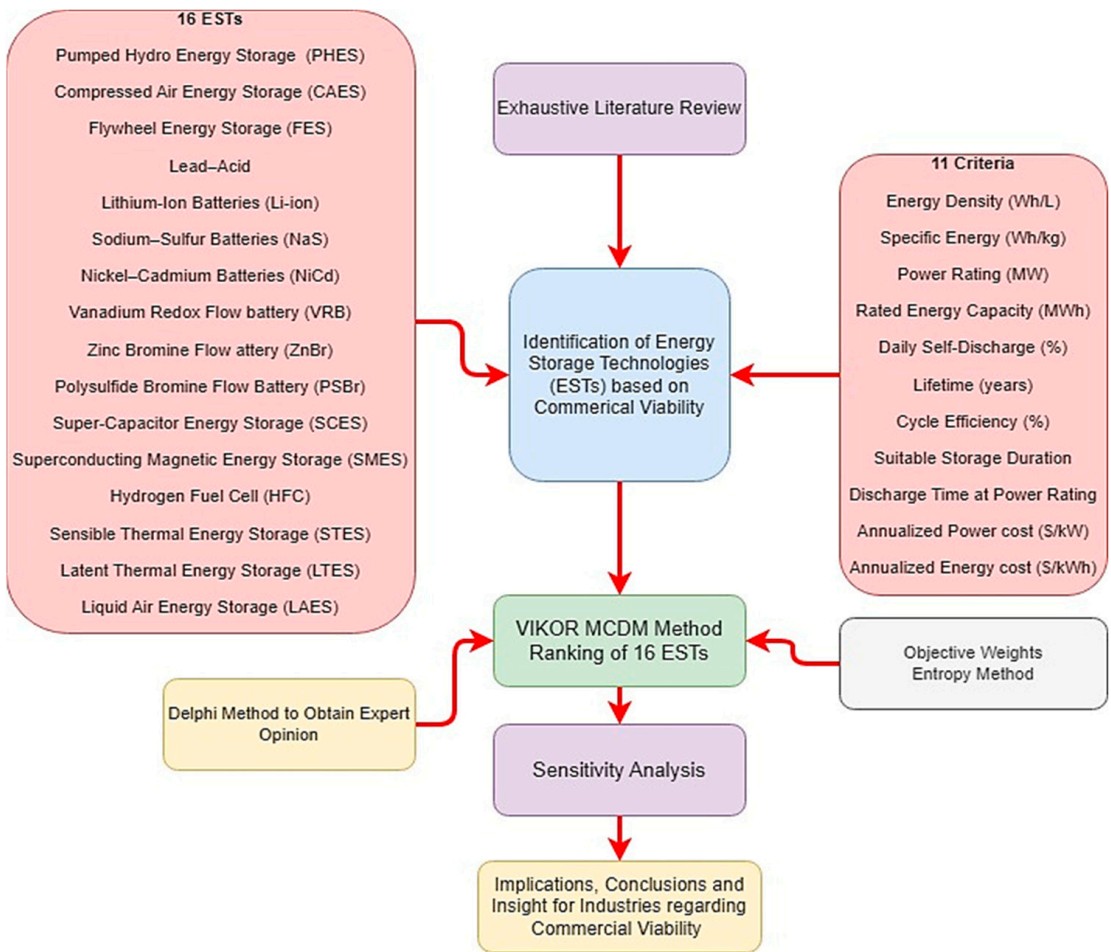

**Figure 1.** Decision making to rank ESTs: Present workflow.

Compressed Air Energy Storage (CAES): CAES compresses air and stocks it in underground grottos. To produce power during peak hours, this air is released into a gas turbine combustor [37].

Flywheel Energy Storage (FES): High-speed mechanical devices called flywheels store electrical energy as rotational energy. The flywheel's rotor is decelerated to issue this energy later in brief energy bursts [38].

Lead–acid: Lead–acid batteries typically have lead (Pb) anodes and lead oxide ($PbO_2$) cathodes that are submerged in sulfuric acid ($H_2SO_4$). Each cell is connected in series. The energy density is quite low, though a large current can lead to many applications. The battery emits lead (toxic heavy metal), posing potential risks to human health with severe impacts on global bioaccumulation. However, lead–acid batteries can be recycled and disposed of successfully, making them economically and environmentally viable [39].

Lithium–ion batteries (Li-ion): In their most prevalent configuration, lithium–ion batteries (Li-ion) are made up of a negative electrode (anode) made of graphite, a positive electrode (cathode) made of lithium oxides, and an electrolyte made of a lithium salt and an organic solvent [39].

Sodium–sulfur batteries (NaS): Molten sodium (Na) serves as the negative electrode in sodium sulphur batteries (NaS), while molten sulfur (S) serves as the positive electrode. Sodium alumina (solid ceramic) is used as an electrolyte. Both sodium and sulphur are cheap and have low densities. Compared to many other batteries, the NaS battery has a high specific energy, a long cycle lifetime, and a high charge efficiency [39]. NaS batteries are a desirable contender for large-scale ES applications because of these benefits [40].

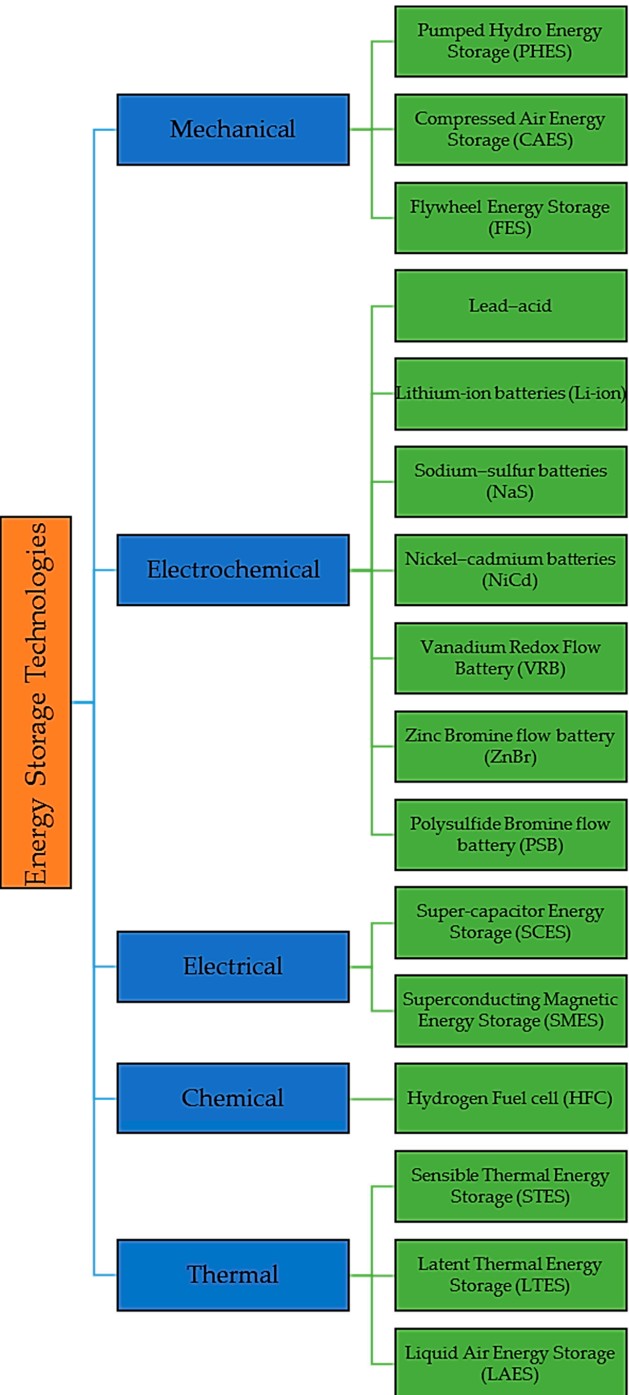

**Figure 2.** Energy Storage Technologies.

Nickel–cadmium batteries (NiCd): NiCd batteries are a type of rechargeable battery and, a reliable power source for portable electronic devices, power tools, and uninterruptible power supplies [6].

Vanadium Redox flow battery (VRB): a VRB is a flow battery that uses a vanadium electrolyte solution to store and release energy. In a VRB, energy is stored in two separate tanks, one containing a positive vanadium electrolyte solution and the other containing a negative vanadium electrolyte solution. When energy is needed, the two solutions are pumped into a battery cell where they interact, generating electricity through an electrochemical reaction. They have a large energy capacity and can store energy for

extended periods, making them appropriate for long-duration ES applications. VRBs have a relatively long life cycle, which can be recharged and used repeatedly for years [6].

Zinc Bromine flow battery (ZnBr): These flow batteries use a zinc–bromine electrolyte solution to store and release energy. In a ZnBr, energy is stored in two separate tanks, one containing a zinc-based positive electrolyte solution and the other containing a bromine-based negative electrolyte solution. When energy is needed, the two solutions are pumped into a battery cell where they interact, generating electricity through an electrochemical reaction. They have a large energy capacity and can store energy for extended periods, making them suitable for use in long-duration ES applications [41].

Polysulfide Bromine flow battery (PSBr): Sodium bromide and sodium polysulphide are used as salt solution electrolytes in a PSBr system. The components utilized in the PSBr system electrolytes are readily available, highly soluble as aqueous electrolytes, and reasonably priced [41].

Super-capacitor Energy Storage (SCES): It stores electrical energy in an electrostatic field. Unlike batteries, which store energy through chemical reactions, supercapacitors store energy by accumulating electric charge on the surface of electrodes. They have higher power density than batteries and can quickly charge and discharge, making them suitable for high-power delivery applications, such as regenerative braking in electric vehicles, backup power, and pulse power applications. However, they have lower energy density than batteries, meaning they cannot store as much energy as batteries of the same size [41].

Superconducting Magnetic Energy Storage (SMES): SMES systems can provide high power output over a short period, making them well-suited for applications such as grid stabilization, frequency regulation, and peak shaving. They also have high energy efficiency, meaning they lose very little energy during storage and release. However, SMES systems require cryogenic cooling to maintain the coils' superconductivity, making them more complex and expensive to maintain compared to other ES systems [42,43].

Hydrogen Fuel cell (HFC): A fuel cell can convert the chemical energy stored in hydrogen and oxygen to electrical energy using the reaction $2H_2 + O_2 \rightarrow 2H_2O +$ energy. The energy generated varies from a few kW to hundreds of MW. The enormous energy generation capacity of HFC is promising; however, disposal/recycling of materials used in fuel cells is a challenge [42,43].

Sensible Thermal Energy Storage (STES): STES is a proven technology that involves cooling or heating a liquid or solid storage medium to store thermal energy. The storage is determined by the material's temperature differential. The capacity of the technique is constrained by the storage medium's specific heat [44].

Latent Thermal Energy Storage (LTES): LTES is the method of stowing thermal energy in a phase-change substance (PCM). The phase change temperature is the temperature at which PCM changes phase (PCT). Technically speaking, the PCM's chemical bonds will begin to disintegrate as the temperature rises above the PCT. The material also transforms from solid to liquid at that time as a result of the material's endothermic reaction to heat. The material will transform back into a solid when the temperature drops because the PCM desorbs heat in an exothermic reaction. It is feasible to store the energy required to change the phase of the material by regulating the temperature within a predetermined rate [45].

Liquid Air Energy Storage (LAES): LAES uses the principles of cryogenics to store and release electrical energy. The system works by compressing air to a liquid state and storing it in a low-pressure container. When electrical energy is needed, the liquid air is vaporized and expanded through a turbine, generating electricity. The resulting high-pressure air can then be recompressed and stored for later use. LAES systems have several advantages over other ES systems, including low cost and widespread availability of the air. However, they are still in the early stages of development and commercial deployment, and their efficiency can be limited by the efficiency of the air compression and expansion process [46]. Table 1 depicts ESTs and their properties [41,47,48].

**Table 1.** ESTs and their properties.

| | Technology | Energy Density (W h/L) | Specific Energy (W h/kg) | Power Rating (MW) | Rated Energy Capacity (MW h) | Daily Self-Discharge (%) | Lifetime (Years) | Cycle Efficiency (%) | Suitable Storage Duration | Discharge Time at Power Rating | Annualized Power Cost ($/ kW) | Annualized Energy Cost ($/kW h) |
|---|---|---|---|---|---|---|---|---|---|---|---|---|
| Mechanical | PHES | 0.5–2 | 0.5–1.5 | 100–5000 | 500–8000 | Very small | 40–60 | 70–85 | Hours–months, long-term | 1–24 h+ | 2500–4300 | 5–100 |
| | CAES | 3–6 | 30–60 | Up to 300 | ~<1000 | Small | 20–40 | 42–54 | Hours–months, long-term | 1–24 h+ | 400–800 | 2–50 |
| | FES | 1000–2000 | 10–30 | <0.25 | up to 5 | 100 | ~15 | 90–95 | Seconds–minutes, short-term | Up to 8 s | 250–350 | 1000–5000 |
| Electrochemical | Lead–acid | 50–80 | 30–50 | 0–20 [4], | 0.001–40 | 0.1–0.3 | 5–15 | 70–80 | Minutes–days, short-to-med. term | up to 10 h | 300–600 | 200–400 |
| | Li-ion batteries | 200–500 | 75–200 | 0–0.1 | 0.004–10 | 0.1–0.3 | 5–15 | ~90–97 | Minutes–days [4], short-to-med. term | 1–8 h | 1200–4000 | 600–2500 |
| | NaS batteries | 150–250 | 150–240 | <8 | 0.4–244.8 | Almost zero | 10–15 | ~75–90 | Long term | ~1 h | 1000–3000 | 300–500 |
| | NiCd batteries | 60–150 | 50–75 | 0–40 | 6.75 | 0.2–0.6 | 10–20 | ~60–70 | Minutes–days [4], Short and long term | ~1–8 h | 500–1500 | 800–1500 |
| | VRB | 16–33 | 10–30 | ~0.03–3 | <60, 2, 3.6 | Small | 5–10 | 75–85 | Hours–months, Long term | Seconds–24 h+ | 600–1500 | 150–1000 |
| | ZnBr flow battery | 30–60 | 30–50 | 0.05–2 | 0.1–3 | Small | 5–10 | ~65–75 | Hours–months, long term | Seconds–10 h+ | 700–2500 | 150–1000 |
| | PSB | ~20–30 | ~15–30 | 1–15 | Potential up to 120 | Small | 10–15 | ~60–75 | Hours–months, long term | Seconds–10 h+ | 700–2500 | 150–1000 |
| Electrical | Super-capacitor | 10–30 | 2.5–15 | 0–0.3 | 0.0005 | 20–40 | 10–30 | ~90–97 | Seconds–hours, short-term | Milliseconds–1 h | 100–300 | 300–2000 |
| | SMES | 0.2–2.5 | 0.5–5 | 0.1–10 | 0.0008 | 10–15 | 20–30 | ~95–97 | Minutes–hours, short-term | Milliseconds–8 s | 200–300 | 1000–10,000 |
| Chemical | Hydrogen Fuel cell | 500–3000 | 800–10,000 | <50 | 0.312 | Almost zero | 5–15 | ~20–50 | Hours–months | Seconds–24 h+ | 500 | 15 |
| Thermal | STES | 80–120 | 80–120 | 0.001–10 | 0.01–0.05 | 0.05–1 | 10–20 | ~30–60 | Minutes–days | 1–8 h | 200–300 | 20–50 |
| | LTES | 0–200 | 10–250 | 0.001–1 | 0.05–0.15 | 0.05–1 | 10–30+ | 75–90 | hours–weeks | 1–8 h | | |
| | LAES | 18–36 | 214 | 10–200 | 2.5 | Small | 25+ | 55–80 | Long-term | Several hours | 900–1900 | 260–530 |

3.1.2. Description of EST Properties

The properties have been identified through a literature review as these are valuable for consideration of any EST towards its commercial viability. The description of properties of EST are as follows [41,47,48]:

- Energy density (Wh/L): Energy density measures the amount of energy that can be stored in a given volume of an energy storage system, usually expressed in watt-hours per litre (Wh/L). A higher energy density means that more energy can be stored in a smaller space, which is important for applications where space is limited.
- Specific energy (Wh/kg): Specific energy measures the amount of energy that can be stored in a given mass of an energy storage system, usually expressed in watt-hours per kilogram (Wh/kg). Higher specific energy means more energy can be stored in a smaller mass, which is important for applications where weight is a concern.
- Power rating (MW): Power rating measures the maximum power an energy storage system can deliver at any given time, usually expressed in megawatts (MW). This is important for high-power output applications like grid-scale energy storage systems.
- Rated energy capacity (MWh): Rated energy capacity measures the maximum amount of energy an energy storage system can store, usually expressed in megawatt-hours (MWh). This is important for large energy storage applications, such as grid-scale energy storage systems.
- Daily self-discharge (%): Daily self-discharge measures the rate at which an energy storage system loses energy over time, expressed as a percentage of its total energy capacity. A lower self-discharge rate means an energy storage system can retain more stored energy over time.
- Lifetime (years): Lifetime is a measure of the expected lifespan of an energy storage system, usually expressed in years. This is an important consideration for applications where the cost of replacing or maintaining an energy storage system is high.
- Cycle efficiency (%): Cycle efficiency measures the efficiency with which an energy storage system can convert stored energy into usable energy and back again, expressed as a percentage. A higher cycle efficiency means less energy is lost during charging and discharging.
- Suitable storage duration: Suitable storage duration measures how long an energy storage system can store energy before recharging. This is an important consideration for applications where energy needs to be stored for an extended period, such as in off-grid energy systems.
- Discharge time at power rating: Discharge time at power rating measures how long an energy storage system can deliver power at its maximum power rating before recharging. This is an important consideration for applications requiring a high power output for a sustained period.
- Annualized Power cost ($/kW): Annualized power cost is a measure of the cost of the energy storage system per unit of power output per year, expressed in dollars per kilowatt (kW). This is an important consideration for applications where the cost of the energy storage system needs to be amortized over its expected lifespan.
- Annualized Energy cost ($/kWh): Annualized energy cost is a measure of the cost of the energy storage system per unit of energy stored per year, expressed in dollars per kilowatt-hour (kWh). This is an important consideration for applications where the cost of the energy storage system needs to be amortized over its expected lifespan.

ESTs play a critical role in achieving a sustainable energy future. They enable energy to be stored and dispatched when it is needed, improving the reliability, resilience, and efficiency of energy systems. MCDM methods have been developed to help assess and compare the different ESTs.

*3.2. Methods Used*

3.2.1. VIKOR Technique

The basic idea of the VIKOR technique for providing a compromise solution to a problem having conflicting criteria was given by S. Opricovic in his PhD dissertation in 1979 with a related paper published in 1980 [49]. The name VIKOR (Serbian: VIseKriterijumska Optimizacija i Kompromisno Resenje, meaning Multicriteria Optimization and Compromise Solution) appeared in 1990 [50]. Recognition of the VIKOR technique came with its comparative analysis to other existing techniques [51].

The VIKOR method involves identifying the alternatives, defining the criteria, determining the weights, and normalizing the criteria values. Hence, they are on a common scale, typically between 0 and 1, then selecting the best and worst values for each criterion among all alternatives. Calculate the S-values for each alternative, which measure its relative performance concerning the best and worst values. Calculate the R-values for each choice, which measure its relative performance concerning the other options. Calculate the Q-values for each alternative, which measure its overall performance based on its S-values and R-values [52]. Rank the alternatives based on their Q-values, with the highest being the best alternative. The VIKOR method is beneficial when conflicting criteria or compromise solutions are needed. It allows decision-makers to identify the best compromise solution that minimizes the distance to the ideal solution while considering the trade-offs between the criteria [53]. The VIKOR technique has the main steps to attain the final ranks of the alternatives as below [54].

Step 1: Identification of the conflicting criteria's $C_j$ where j = 1, 2, 3, . . . , n; and the alternatives $A_i$ where i = 1, 2, 3, . . . , m for analysis and ranking. Create the decision matrix X with $I_{ij}$ as respective values based on $C_j$ and $A_i$; refer to Equation (1).

$$X = \begin{matrix} & \begin{matrix} C_1 & C_2 & \dots & C_n \end{matrix} \\ \begin{matrix} A_1 \\ A_2 \\ \dots \\ A_m \end{matrix} & \begin{bmatrix} I_{11} & I_{21} & \dots & I_{1n} \\ I_{21} & I_{22} & \dots & I_{2n} \\ \dots & \dots & \dots & \dots \\ I_{m1} & I_{m2} & \dots & I_{mn} \end{bmatrix} \end{matrix} \tag{1}$$

Identify each criterion ($C_j$) as beneficial (B) or non-beneficial (NB). A criterion is beneficial if an increment in its value is sought, while a criterion is non-beneficial if a decrement in its value is sought.

Step 2: Evaluate the importance of each criterion ($C_j$). There are several methods for calculating weights, including the Pairwise Comparison method of the AHP [55], the entropy method [56], the Best-Worst method [57], and the Full Consistency method [58,59].

Step 3: Regulate the normalized decision matrix where each element can be calculated per Equation (2).

$$f_{ij} = \frac{I_{ij}}{\sqrt{\sum_{i=1}^{m} (I_{ij})^2}} \tag{2}$$

where i = 1, 2, 3, . . . , m and j = 1, 2, 3, . . . , n

Step 4: Determine the best $f_j^+$ and the worst $f_j^-$ values of all criterion, j = 1, 2, . . . , n;
$f_j^+$ = max ($f_j$, j = 1, 2, 3, . . . , n) for each criterion.
$f_j^-$ = min ($f_j$, j = 1, 2, 3, . . . , n) for each criterion.

Based on the above best ($f_j^+$) and worst ($f_j^-$) value of each criterion, calculate the maximum gap of improvement for beneficial and non-beneficial criteria as per Equations (3) and (4), respectively.

$$S_{ij} = w_j \left[ \frac{f_j^+ - f_{ij}}{f_j^+ - f_j^-} \right] \tag{3}$$

$$S_{ij} = w_j \left[ \frac{f_{ij} - f_j^-}{f_j^+ - f_j^-} \right] \tag{4}$$

Step 5: Calculate the Utility measure ($S_i$) Equation (5) and Regret measure ($R_i$) Equation (6) for each alternative.

$$S_i = \sum_{j=1}^{n} S_{ij} \tag{5}$$

$$R_i = max\left[S_{ij}\right] \, for \, j = 1, 2, \ldots n \tag{6}$$

Step 6: Calculate the VIKOR index ($Q_i$) for each alternative as per Equation (7).

$$Q_i = v\left[\frac{S_i - S_i^-}{S_i^+ - S_i^-}\right] + (1 - v)\left[\frac{R_i - R_i^-}{R_i^+ - R_i^-}\right] \tag{7}$$

where, $S_i^+$ is the maximum value of $S_i$ and $S_i^-$ is the minimum value of $S_i$

$R_i^+$ is the maximum value of $R_i$ and $R_i^-$ is the minimum value of $R_i$

$v$ is the weight for the strategy of decision-making group utility. The value of $v$ ranges from 0 to 1, though it is usually set to 0.5 by consensus.

Step 7: Rank the alternatives, in order of lowest to highest values of $S_i$, $R_i$, and $Q_i$ based on fulfilling the following two conditions.

Condition 1—Acceptable advantage: $Q_{(Rank2)} - Q_{(Rank1)} \geq (1/(n - 1))$, where $Q_{(Rank2)}$ and $Q_{(Rank1)}$ are the value of alternatives occupying the second and first position, respectively, in the ranking list by Q; n is the number of alternatives.

Condition 2—Acceptable stability in decision making: The alternative ranked first by $Q_i$ must also be ranked best by {$S_i$ or/and $R_i$ | i = 1, 2, . . . , m}.

If one of the conditions is not satisfied, then a set of compromise solutions is proposed:

- Alternatives ranked first and second if only condition 2 is not satisfied.
- Alternatives ranked 1 to k, if condition 1 is not satisfied. The relation of condition 1 determines rank k.

### 3.2.2. Entropy Method

The entropy method is used in decision-making to determine the weight or importance of different criteria or attributes. It is based on information theory and the idea that the more uncertain or unpredictable a criterion is, the more information it provides. It involves calculating the entropy value for each criterion, representing the degree of uncertainty or diversity in the data. A criterion with a high entropy value indicates a wide range of values or options, while a low entropy value means that the data is more concentrated or uniform. Once the entropy values are calculated, they are used to determine the weight or importance of each criterion. The weight of each criterion is proportional to its entropy value, with the criterion having the highest entropy value assigned the highest weight [56].

S1: Normalize the decision matrix using Equation (8).

$$p_{ij} = \frac{I_{ij}}{\sum_{j=1}^{m} I_{ij}} \tag{8}$$

where $i = 1, \ldots, n; j = 1, \ldots, m$. This normalization technique converts various scales and units into common measurable units to enable comparisons between multiple criteria.

S2: Compute entropy $e_i$ for each criterion as per Equation (9).

$$e_i = -e_0 \sum_{j=1}^{m} p_{ij}. \ln p_{ij} \tag{9}$$

where, $e_0$ is the entropy constant and is equal to $(\ln m)^{-1}$ and $p_{ij}. \ln p_{ij} = 0$ if $p_{ij} = 0$.

Here, $e_0 = (\ln 16)^{-1}$.

S3: Calculate the degree of diversification according to Equation (10).

$$di = 1 - e_i, i = 1, \ldots, n \tag{10}$$

S4: Calculate the weightage of each criterion according to Equation (11).

$$w_i = \frac{d_i}{\sum_{i=1}^{n} d_i}, \ i = 1, \ldots, n \tag{11}$$

### 3.2.3. Delphi Method

The Delphi method is a structured process to obtain and refine expert opinions on a particular topic or issue. It typically involves three rounds of questioning, with feedback provided to the experts after each round. In the first round, the experts are asked to provide their opinions on questions related to the topic or issue. These questions may be open-ended or closed-ended, designed to elicit various opinions and perspectives. In the second round, the experts are provided with a summary of the responses from the first round, along with any statistical analysis or other feedback that may be relevant. They are then asked to revise their initial responses in light of this feedback. In the third round, the experts are again provided with a summary of the responses from the second round, along with any additional feedback that may be relevant. They are then asked to provide their final responses [60].

## 4. Ranking of Energy Storage Technologies

Step 1: Identified criteria are shown in Table 2, and ESTs are shown in Table 3.

**Table 2.** Criteria for analysis of ESTs.

| Designation | Criteria (Unit of Measurement) |
|:---:|:---:|
| $C_1$ | Energy density (Wh/L) |
| $C_2$ | Specific energy (Wh/kg) |
| $C_3$ | Power rating (MW) |
| $C_4$ | Rated energy capacity (MWh) |
| $C_5$ | Daily self-discharge (score) |
| $C_6$ | Lifetime (years) |
| $C_7$ | Cycle efficiency (%) |
| $C_8$ | Suitable storage duration (score) |
| $C_9$ | Discharge time at power rating (score) |
| $C_{10}$ | Annualized Power Cost ($/Kw) |
| $C_{11}$ | Annualized Energy Cost ($/Kwh) |

**Table 3.** Alternatives of ESTs.

| Designation | Alternatives |
|:---:|:---:|
| $A_1$ | PHES |
| $A_2$ | CAES |
| $A_3$ | FES |
| $A_4$ | Lead–acid |
| $A_5$ | Lithium-ion batteries |
| $A_6$ | NaS batteries |
| $A_7$ | NiCd batteries |
| $A_8$ | VRB |
| $A_9$ | ZnBr |
| $A_{10}$ | PSBr |
| $A_{11}$ | SCES |
| $A_{12}$ | SMES |
| $A_{13}$ | HFC |
| $A_{14}$ | STES |
| $A_{15}$ | LTES |
| $A_{16}$ | LAES |

Decision Matrix (X) is created by considering data available from Table 1 and using the Delphi method [61]. The final data are agreed by five academic experts on EST and is compiled in Table 4.

**Table 4.** Decision matrix.

|  | $C_1$ | $C_2$ | $C_3$ | $C_4$ | $C_5$ | $C_6$ | $C_7$ | $C_8$ | $C_9$ | $C_{10}$ | $C_{11}$ |
|---|---|---|---|---|---|---|---|---|---|---|---|
| $A_1$ | 2 | 1.5 | 4000 | 500 | 1 | 60 | 85 | 4 | 2 | 308 | 19 |
| $A_2$ | 6 | 60 | 300 | 580 | 1 | 40 | 50 | 4 | 2 | 203 | 13 |
| $A_3$ | 2000 | 30 | 20 | 0.75 | 5 | 15 | 95 | 1 | 1 | 293 | 3069 |
| $A_4$ | 80 | 50 | 20 | 0.001 | 1 | 15 | 80 | 2 | 2 | 866 | 216 |
| $A_5$ | 500 | 200 | 50 | 0.024 | 1 | 15 | 90 | 2 | 2 | 312 | 78 |
| $A_6$ | 250 | 240 | 8 | 0.4 | 1 | 15 | 90 | 5 | 2 | 490 | 123 |
| $A_7$ | 150 | 75 | 40 | 6.75 | 1 | 20 | 70 | 2 | 2 | 1500 | 2400 |
| $A_8$ | 33 | 30 | 3 | 2 | 1 | 10 | 85 | 4 | 2 | 464 | 116 |
| $A_9$ | 60 | 50 | 2 | 3 | 1 | 10 | 75 | 4 | 2 | 365 | 91 |
| $A_{10}$ | 30 | 30 | 15 | 0.06 | 1 | 15 | 75 | 4 | 2 | 700 | 150 |
| $A_{11}$ | 30 | 15 | 0.3 | 0.0005 | 3 | 10 | 90 | 1 | 1 | 109 | 14,880 |
| $A_{12}$ | 2.5 | 5 | 10 | 0.015 | 3 | 20 | 95 | 1 | 1 | 300 | 10,000 |
| $A_{13}$ | 3000 | 10,000 | 50 | 0.3 | 1 | 15 | 50 | 4 | 2 | 500 | 20 |
| $A_{14}$ | 500 | 250 | 10 | 0.05 | 1 | 20 | 60 | 2 | 2 | 300 | 150 |
| $A_{15}$ | 200 | 250 | 1 | 0.15 | 1 | 20 | 80 | 3 | 2 | 400 | 180 |
| $A_{16}$ | 30 | 214 | 200 | 2.5 | 1 | 25 | 70 | 5 | 2 | 1900 | 530 |
|  | B | B | B | B | B | B | B | B | B | NB | NB |

Wherein, the last row of the decision matrix represents the criteria as beneficial (B) or non-beneficial (NB).

Step 2: The weightage of each criterion is calculated using the entropy method consisting of steps S1 to S4.

Based on the above entropy calculations, the weightage of each criterion is shown in Table 5.

**Table 5.** Weightage of the criteria.

| $C_1$ | $C_2$ | $C_3$ | $C_4$ | $C_5$ | $C_6$ | $C_7$ | $C_8$ | $C_9$ | $C_{10}$ | $C_{11}$ |
|---|---|---|---|---|---|---|---|---|---|---|
| 0.069 | 0.148 | 0.137 | 0.12 | 0.075 | 0.066 | 0.061 | 0.061 | 0.06 | 0.071 | 0.132 |

Step 3: The decision matrix in Table 4 is normalized as shown in Table 6.

**Table 6.** Normalized Decision matrix.

|  | $C_1$ | $C_2$ | $C_3$ | $C_4$ | $C_5$ | $C_6$ | $C_7$ | $C_8$ | $C_9$ | $C_{10}$ | $C_{11}$ |
|---|---|---|---|---|---|---|---|---|---|---|---|
| $A_1$ | 0.0005 | 0.0001 | 0.9957 | 0.6529 | 0.1336 | 0.6298 | 0.2698 | 0.3032 | 0.2697 | 0.1050 | 0.0010 |
| $A_2$ | 0.0016 | 0.0060 | 0.0747 | 0.7574 | 0.1336 | 0.4199 | 0.1587 | 0.3032 | 0.2697 | 0.0692 | 0.0007 |
| $A_3$ | 0.5416 | 0.0030 | 0.0050 | 0.0010 | 0.6682 | 0.1575 | 0.3015 | 0.0758 | 0.1348 | 0.0999 | 0.1672 |
| $A_4$ | 0.0217 | 0.0050 | 0.0050 | 0.0000 | 0.1336 | 0.1575 | 0.2539 | 0.1516 | 0.2697 | 0.2951 | 0.0118 |
| $A_5$ | 0.1354 | 0.0200 | 0.0124 | 0.0000 | 0.1336 | 0.1575 | 0.2857 | 0.1516 | 0.2697 | 0.1063 | 0.0042 |
| $A_6$ | 0.0677 | 0.0240 | 0.0020 | 0.0005 | 0.1336 | 0.1575 | 0.2857 | 0.3790 | 0.2697 | 0.1670 | 0.0067 |
| $A_7$ | 0.0406 | 0.0075 | 0.0100 | 0.0088 | 0.1336 | 0.2099 | 0.2222 | 0.1516 | 0.2697 | 0.5112 | 0.1307 |
| $A_8$ | 0.0089 | 0.0030 | 0.0007 | 0.0026 | 0.1336 | 0.1050 | 0.2698 | 0.3032 | 0.2697 | 0.1581 | 0.0063 |
| $A_9$ | 0.0162 | 0.0050 | 0.0005 | 0.0039 | 0.1336 | 0.1050 | 0.2381 | 0.3032 | 0.2697 | 0.1244 | 0.0050 |
| $A_{10}$ | 0.0081 | 0.0030 | 0.0037 | 0.0001 | 0.1336 | 0.1575 | 0.2381 | 0.3032 | 0.2697 | 0.2386 | 0.0082 |
| $A_{11}$ | 0.0081 | 0.0015 | 0.0001 | 0.0000 | 0.4009 | 0.1050 | 0.2857 | 0.0758 | 0.1348 | 0.0371 | 0.8105 |
| $A_{12}$ | 0.0007 | 0.0005 | 0.0025 | 0.0000 | 0.4009 | 0.2099 | 0.3015 | 0.0758 | 0.1348 | 0.1022 | 0.5447 |
| $A_{13}$ | 0.8123 | 0.9986 | 0.0124 | 0.0004 | 0.1336 | 0.1575 | 0.1587 | 0.3032 | 0.2697 | 0.1704 | 0.0011 |
| $A_{14}$ | 0.1354 | 0.0250 | 0.0025 | 0.0001 | 0.1336 | 0.2099 | 0.1905 | 0.1516 | 0.2697 | 0.1022 | 0.0082 |
| $A_{15}$ | 0.0542 | 0.0250 | 0.0002 | 0.0002 | 0.1336 | 0.2099 | 0.2539 | 0.2274 | 0.2697 | 0.1363 | 0.0098 |
| $A_{16}$ | 0.0081 | 0.0214 | 0.0498 | 0.0033 | 0.1336 | 0.2624 | 0.2222 | 0.3790 | 0.2697 | 0.6475 | 0.0289 |

Step 4: Best and worst value of each criterion is determined as shown in Table 7.

**Table 7.** Best and worst value of each criterion.

|  | C₁ | C₂ | C₃ | C₄ | C₅ | C₆ | C₇ | C₈ | C₉ | C₁₀ | C₁₁ |
|---|---|---|---|---|---|---|---|---|---|---|---|
| Best $(f_j^+)$ | 0.8123 | 0.9986 | 0.9957 | 0.7574 | 0.6682 | 0.6298 | 0.3015 | 0.3790 | 0.2697 | 0.6475 | 0.8105 |
| Worst $(f_j^-)$ | 0.0005 | 0.0001 | 0.0001 | 0.0000 | 0.1336 | 0.1050 | 0.1587 | 0.0758 | 0.1348 | 0.0371 | 0.0007 |

Based on the above best and worst values of each criterion, the calculation of the maximum gap of improvement for beneficial and non-beneficial criteria has been done as shown in Table 8.

**Table 8.** Improvement gap matrix.

|  | C₁ | C₂ | C₃ | C₄ | C₅ | C₆ | C₇ | C₈ | C₉ | C₁₀ | C₁₁ |
|---|---|---|---|---|---|---|---|---|---|---|---|
| A₁ | 0.0690 | 0.1480 | 0.0000 | 0.0166 | 0.0750 | 0.0000 | 0.0136 | 0.0153 | 0.0000 | 0.0079 | 0.0001 |
| A₂ | 0.0689 | 0.1471 | 0.1267 | 0.0000 | 0.0750 | 0.0264 | 0.0610 | 0.0153 | 0.0000 | 0.0037 | 0.0000 |
| A₃ | 0.0230 | 0.1476 | 0.1363 | 0.1198 | 0.0000 | 0.0594 | 0.0000 | 0.0610 | 0.0600 | 0.0073 | 0.0271 |
| A₄ | 0.0672 | 0.1473 | 0.1363 | 0.1200 | 0.0750 | 0.0594 | 0.0203 | 0.0458 | 0.0000 | 0.0300 | 0.0018 |
| A₅ | 0.0575 | 0.1451 | 0.1353 | 0.1200 | 0.0750 | 0.0594 | 0.0068 | 0.0458 | 0.0000 | 0.0080 | 0.0006 |
| A₆ | 0.0633 | 0.1445 | 0.1367 | 0.1199 | 0.0750 | 0.0594 | 0.0068 | 0.0000 | 0.0000 | 0.0151 | 0.0010 |
| A₇ | 0.0656 | 0.1469 | 0.1356 | 0.1186 | 0.0750 | 0.0528 | 0.0339 | 0.0458 | 0.0000 | 0.0551 | 0.0212 |
| A₈ | 0.0683 | 0.1476 | 0.1369 | 0.1196 | 0.0750 | 0.0660 | 0.0136 | 0.0153 | 0.0000 | 0.0141 | 0.0009 |
| A₉ | 0.0677 | 0.1473 | 0.1369 | 0.1194 | 0.0750 | 0.0660 | 0.0271 | 0.0153 | 0.0000 | 0.0101 | 0.0007 |
| A₁₀ | 0.0684 | 0.1476 | 0.1365 | 0.1200 | 0.0750 | 0.0594 | 0.0271 | 0.0153 | 0.0000 | 0.0234 | 0.0012 |
| A₁₁ | 0.0684 | 0.1478 | 0.1370 | 0.1200 | 0.0375 | 0.0660 | 0.0068 | 0.0610 | 0.0600 | 0.0000 | 0.1320 |
| A₁₂ | 0.0690 | 0.1479 | 0.1367 | 0.1200 | 0.0375 | 0.0528 | 0.0000 | 0.0610 | 0.0600 | 0.0076 | 0.0887 |
| A₁₃ | 0.0000 | 0.0000 | 0.1353 | 0.1199 | 0.0750 | 0.0594 | 0.0610 | 0.0153 | 0.0000 | 0.0155 | 0.0001 |
| A₁₄ | 0.0575 | 0.1443 | 0.1367 | 0.1200 | 0.0750 | 0.0528 | 0.0474 | 0.0458 | 0.0000 | 0.0076 | 0.0012 |
| A₁₅ | 0.0644 | 0.1443 | 0.1370 | 0.1200 | 0.0750 | 0.0528 | 0.0203 | 0.0305 | 0.0000 | 0.0115 | 0.0015 |
| A₁₆ | 0.0684 | 0.1449 | 0.1302 | 0.1195 | 0.0750 | 0.0462 | 0.0339 | 0.0000 | 0.0000 | 0.0710 | 0.0046 |

Step 5: Utility measure ($S_i$) and regret measure ($R_i$) can be easily calculated using the improvement gap matrix which is shown in Table 9.

**Table 9.** Utility and Regret Measure.

|  | Utility Measure ($S_i$) | Regret Measure ($R_i$) |
|---|---|---|
| A1 | 0.3453 | 0.148 |
| A2 | 0.5242 | 0.1471 |
| A3 | 0.6416 | 0.1476 |
| A4 | 0.7031 | 0.1473 |
| A5 | 0.6534 | 0.1451 |
| A6 | 0.6217 | 0.1445 |
| A7 | 0.7505 | 0.1469 |
| A8 | 0.6572 | 0.1476 |
| A9 | 0.6655 | 0.1473 |
| A10 | 0.6738 | 0.1476 |
| A11 | 0.8364 | 0.1478 |
| A12 | 0.7811 | 0.1479 |
| A13 | 0.4814 | 0.1353 |
| A14 | 0.6883 | 0.1443 |
| A15 | 0.6574 | 0.1443 |
| A16 | 0.6935 | 0.1449 |

Step 6: VIKOR index (Qi) can be calculated using v = 0.5 and is shown in Table 10.

**Table 10.** VIKOR Index ($Q_i$).

| | VIKOR Index ($Q_i$) |
|---|---|
| $A_1$ | 0.5 |
| $A_2$ | 0.648 |
| $A_3$ | 0.785 |
| $A_4$ | 0.836 |
| $A_5$ | 0.6981 |
| $A_6$ | 0.6424 |
| $A_7$ | 0.8697 |
| $A_8$ | 0.8009 |
| $A_9$ | 0.7977 |
| $A_{10}$ | 0.8178 |
| $A_{11}$ | 0.9921 |
| $A_{12}$ | 0.9417 |
| $A_{13}$ | 0.1386 |
| $A_{14}$ | 0.7044 |
| $A_{15}$ | 0.6729 |
| $A_{16}$ | 0.7307 |

Step 7. Ranking based on decreasing values of $S_i$, $R_i$, and $Q_i$ is shown in Table 11.

**Table 11.** Ranking of alternatives.

| | $S_i$ | $R_i$ | $Q_i$ |
|---|---|---|---|
| A1 | 1 | 16 | 2 |
| A2 | 3 | 8 | 4 |
| A3 | 5 | 11 | 9 |
| A4 | 13 | 9 | 13 |
| A5 | 6 | 6 | 6 |
| A6 | 4 | 4 | 3 |
| A7 | 14 | 7 | 14 |
| A8 | 7 | 11 | 11 |
| A9 | 9 | 9 | 10 |
| A10 | 10 | 11 | 12 |
| A11 | 16 | 14 | 16 |
| A12 | 15 | 15 | 15 |
| A13 | 2 | 1 | 1 |
| A14 | 11 | 2 | 7 |
| A15 | 8 | 2 | 5 |
| A16 | 12 | 5 | 8 |

Checking for Condition 1—Acceptable advantage: $Q_{(Rank2)} - Q_{(Rank1)} \geq (1/(n-1))$, where $Q_{(Rank2)}$ and $Q_{(Rank1)}$ are the value of alternatives occupying the second and first position respectively in the ranking list by Q; n is the number of alternatives. From Table 10:
0.5000 − 0.1386 ≥ 1/15 or 0.3614 ≥ 0.0667, which is true. Hence, condition 1 is satisfied.

Checking for Condition 2—Acceptable stability in decision making: alternative ranked first by $Q_i$ must also be the best ranked by {$S_i$ or/and $R_i$ | i = 1, 2, . . . , m}.

As we can see from Table 11, alternative $A_{13}$ is ranked 1st by $R_i$. Although, alternative $A_{13}$ is ranked 2nd by $S_i$; condition 2 is satisfied as the alternative should be ranked first either by $S_i$ or $R_i$.

Thus, Alternative $A_{13}$ (Hydrogen Fuel cell (HFC)) of the considered ESTs is the most viable option for commercialization. HFC is a clean and efficient source of energy, with the potential to reduce greenhouse gas emissions and contribute to a low-carbon economy. The commercial viability of hydrogen fuel cells is driven by several factors. Firstly, HFCs have high energy density and long-life cycles, making them suitable for an extensive variety

of applications, from transportation to stationary power. Secondly, the technology has advanced significantly in recent years, resulting in reduced costs and increased reliability, making them more accessible to a wider range of customers. Furthermore, the growing demand for clean energy and the need for sustainable ES solutions is driving the development and commercialization of hydrogen fuel cells. Governments and businesses around the world are investing in hydrogen fuel cell research and development, and the infrastructure needed to support hydrogen fuel cell use, such as hydrogen production and distribution, is also being developed.

The VIKOR method has assigned a first rank to Hydrogen Fuel Cells, second to PHES, third to Sodium–Sulfur Batteries, and fourth to Lead–Acid batteries. This ranking reflects the relative strengths and weaknesses of these ESTs, as evaluated against a set of pre-determined criteria such as cost, energy efficiency, safety, environmental impact, scalability, etc. Hydrogen fuel cells have been assigned the first rank due to their high energy density, long life cycle, and clean and efficient operation. Additionally, the expansion of hydrogen fuel cell technology and the infrastructure needed to support it are advancing rapidly, making hydrogen fuel cells an attractive option for commercialization. PHES has been assigned the second rank due to its low cost, high energy density, and proven reliability. However, the construction of PHES systems requires a significant amount of infrastructure, including reservoirs and pipelines, and their deployment is limited by geographical constraints. Sodium–Sulfur batteries have been assigned the third rank due to their high energy density and efficiency. However, the high cost of sodium-sulfur batteries and their relatively short life cycle compared to other ES technologies have limited their commercial viability. Lead–Acid batteries have been assigned the fourth rank due to their low cost, proven reliability, and widespread availability. However, lead–acid batteries have relatively low energy density and short life cycles, and the production of lead and sulfuric acid used in the batteries can have negative environmental impacts. The implementation of MCDM has provided valuable insights into the strengths and weaknesses of different ES technologies, and the ranking assigned to each technology can help decision-makers to make informed choices about which technology to adopt.

## 5. Sensitivity Analysis

This can provide insights into the results' stability and help decision-makers identify which criteria have the greatest impact on the final rankings. Two types of sensitivity analysis exist: a one-at-a-time and a global sensitivity analysis. In one-at-a-time sensitivity analysis, the impact of a change in each criterion weight is evaluated individually, while in global sensitivity analysis, the impact of simultaneous changes in multiple criteria weights is evaluated.

In calculating the VIKOR Index ($Q_i$), $v$ is the weight for the decision-making strategy, whose value ranges from 0 to 1. The weight for the decision-making strategy by consensus is taken as 0.5. However, $v > 0.5$ is considered using "voting by majority rule", while $v < 0.5$ is taken with "veto".

Sensitivity analysis was done to ascertain the position of targeted alternative $A_{13}$ (HFC) and other alternatives against different values of $v$. The values of $v$ have been taken from 0 to 1 with a gap of 0.1. The values have been plotted to visualize their position, as shown in Figure 3.

### 5.1. Implications of the Study

MCDM is crucial in analyzing different ESTs based on identified energy recital indicators. MCDM is a systematic process that considers multiple criteria, such as cost, performance, sustainability, and commercial viability, to evaluate and rank ES technologies. It enables decision-makers to identify the best option among alternatives by considering the trade-offs between the different criteria. Applying the VIKOR method provided a structured and systematic approach to evaluating and ranking ESTs, reducing the chances of suboptimal or irrational decisions. It helped to ensure transparency in the decision-

making process, which can increase stakeholder confidence in the results and decisions. MCDM enables decision-makers to consider and understand the trade-offs between energy performance indicators, such as cost, performance, and sustainability. By evaluating ES technologies based on multiple criteria, MCDM can help to identify the best option for a given situation, potentially improving competitiveness in the ES industry. MCDM can increase stakeholder engagement by involving them in decision-making, leading to better alignment and more informed decision-making. Sensitivity analysis helped regulate the stability and robustness of the results and reduced the chances of reaching suboptimal decisions. The decision-makers can better understand the alternatives' strengths and weaknesses and make more informed decisions.

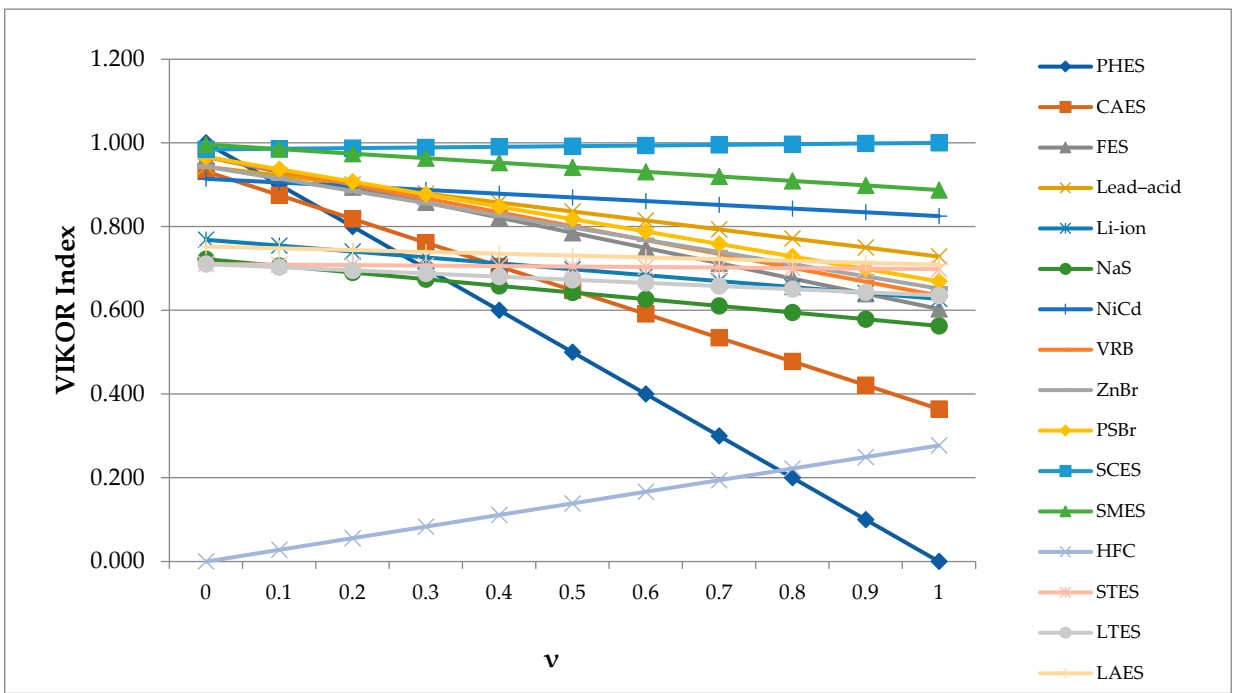

**Figure 3.** Sensitivity analysis plot–VIKOR index vs. *v*.

*5.2. Theoretical Contributions*

The study comprehensively analyses various energy ESTs based on their commercialization viability. ESTs are essential for ensuring that energy is available when required, and many different technologies are available, including mechanical, electrochemical, electrical, chemical, and thermal. However, not all ESTs are commercially viable, and this study aims to identify which technologies are most likely to be successful in the market. It uses the VIKOR technique to analyze ESTs, ranks them according to their commercialization potential, and highlights the importance of commercialization viability in making an eco-friendly and optimal solution for storing energy for a longer duration. This study integrates the Entropy weights method with the VIKOR technique to assign objective weights to the identified energy performance indicators. The procedure helps a consumer choose better ESTs as per their requirement while manufacturers compete with each other to enhance the commercial value of their energy storage products. The sensitivity analysis helps to understand the uncertainties, pros, and cons, along with the limitations and scope of using the decision model. The study of different ESTs indicates that HFCs are impressive and promising for the future. This study identifies HFC as a technology with high commercialization viability, making it an attractive option for energy storage. The findings provide insights into the potential of HFCs and could help policymakers and industry leaders make informed decisions regarding energy storage technologies.

## 6. Conclusions

Energy Storage Technologies (EST) are required to fulfil the ever-increasing demand for reliable power sources by industries, households, and others. A few ESTs are already matured ones with extensive use, like PHES, Lead–Acid, VRB, etc. Though, there are certain limitations. They do not fit in every power usage scenario. Newer ESTs are being invented and developed wherein Hydrogen Fuel Cell (HFC) seems impressive and promising for the future as the VIKOR method assigned its first rank. It fits almost all types of power usage, like commercial, industrial, residential, or remote areas. The best future energy answer may be hydrogen, but getting there will take political will and financial commitment. However, when fossil fuels become scarce, hydrogen might be a significant answer to our world's energy problems.

The evaluation model of EST has been developed, keeping in view their commercial viability. Entropy and VIKOR techniques have helped make an informed decision on choosing the better EST.

Hydrogen fuel cells are a highly viable option for commercialization among ESTs, with a combination of high energy density, long life cycles, reduced costs, and growing demand for clean energy, all contributing to their commercial viability. By continuing to invest in research and development, the hydrogen fuel cell industry can significantly contribute to a low-carbon, sustainable future.

Hydrogen fuel cells are a viable option for commercialization for several reasons:

- Environmentally Friendly: Hydrogen fuel cells emit only water and heat, making them a clean and renewable energy source. This makes them a promising alternative to traditional fossil fuels emitting harmful greenhouse gases and contribute to climate change.
- High Efficiency: Fuel cells have a higher efficiency rate than traditional combustion engines, which can convert more fuel energy into usable power. This can result in greater fuel economy and reduced operating costs.
- Versatile Applications: Hydrogen fuel cells can be used in various applications, including transportation, power generation, and energy storage. This versatility makes them an attractive option for commercialization.
- Decreasing Costs: The cost of producing hydrogen fuel cells has steadily declined over the years, making them more affordable and accessible commercially.
- Government Support: Many governments worldwide are investing in developing and commercializing hydrogen fuel cells to reduce greenhouse gas emissions and transition to a cleaner energy future. This support can help to drive innovation and accelerate the adoption of fuel cells in commercial applications.

The commercial viability of hydrogen fuel cells is still a work in progress, but the potential benefits of this technology make it an attractive option for commercialization. With continued investment and collaboration, overcoming the challenges and creating a sustainable and healthy power source for a wide range of applications is possible. Despite hydrogen fuel cells being one of the best renewable energy sources, some challenges must be overcome before hydrogen can completely fulfil its possibilities as a solution by providing a prospective carbon-free energy system.

Limitations of the research are the values of criteria taken through different research papers and the data are not concurrent. In future, concurrent experimental data may be collected, and techniques may be used for a more realistic result.

Overall, the combination of their environmental benefits, high efficiency, versatility, decreasing costs, and government support make hydrogen fuel cells a viable option for commercialization. As research and development continue, we may see more widespread adoption of this promising technology.

**Author Contributions:** Conceptualization, X.S., R.K., R.K.S., N.D., Ž.S. and S.S.; methodology, R.K.S., N.D., Ž.S., S.S. and M.R.; software, R.K.S., N.D., Ž.S. and S.S.; validation, X.S., R.K., R.K.S., S.S. and M.R.; formal analysis, X.S., R.K., R.K.S., N.D. and Ž.S.; investigation, X.S., R.K., R.K.S., N.D., Ž.S., S.S. and M.R.; resources, R.K., R.K.S., N.D. and M.R.; data curation, R.K.S. and N.D.; writing—original draft preparation, R.K.S., N.D. and S.S.; writing—review and editing, R.K., Ž.S., S.S. and M.R.; visualization, X.S., R.K., R.K.S., N.D., Ž.S., S.S. and M.R.; supervision, X.S., Ž.S., S.S. and M.R.; project administration, X.S., R.K. and Ž.S.; funding acquisition, X.S. and Ž.S. All authors have read and agreed to the published version of the manuscript.

**Funding:** This research received no external funding.

**Institutional Review Board Statement:** Not applicable.

**Informed Consent Statement:** Not applicable.

**Data Availability Statement:** Data are within this study.

**Conflicts of Interest:** The authors declare no conflict of interest.

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
