# Peer review of "Sustainability Assessment of Energy Storage Technologies Based on Commercialization Viability: MCDM Model"

_sustainability, doi:10.3390/su15064707_

Round 1

Reviewer 1 Report

1- why do you only use VIKOR and ENTROPY , Why not other methods such as AHP , ANP or TOPSIS

2- The theoretical contributions of the research should be addressed.

3. There lacks of literature review in terms of MCDM theories and applications. 4. The discussions and managerial insights are necessary.

Author Response

Sr. No.

Comments

Corrective Action and Response

1

Why do you only use VIKOR and ENTROPY, Why not other methods such as AHP, ANP or TOPSIS

The criteria weights can be calculated using the entropy method in VIKOR and Entropy, which uses the values of the criteria itself. In the case of AHP/ANP, the pairwise comparison is made, which deals with the human factor and may result in biasing. Even in TOPSIS, generally, weights from AHP are used. Thus, VIKOR and entropy method is more rational in finding the weight of criteria and hence used.

2

The theoretical contributions of the research should be addressed.

Section 5.2 theoretical contributions is added.

3

There lacks of literature review in terms of MCDM theories and applications.

A literature review on MCDM has been added.

4

The discussions and managerial insights are necessary.

The same has been added in the conclusion section.

Reviewer 2 Report

The manuscript belongs to the topics of the journal and deals with important problems. The authors have applied Multi-Criteria Decision-Making methods to make a sustainable assessment of energy storage technologies based on commercialization Viability. Generally, the manuscript has core scientific elements needed to be considered in this journal. Currently, the quality of the manuscript allows being revised according to the next suggestions. - Language in the manuscript in a few places needs to be improved. - The abstract missing some core elements. Please add novelty, contribution, and results in the short term. - Overall structure of the paper isn't appropriate. The introduction is too long and contains one Figure and one Table. Must be restructured in the following way: In the introduction, you should add clear goals of the research, contributions, and reasons for writing this study. The second section should be a Literature review (Background), and many current introductions can be moved into this section. The third section can be Materials and methods. In this part of the manuscript should be moved Figure 1, and Table 1. Also, the text beginning from "Pumped Hydro Energy Storage (PHES): PHES works by using excess electricity gener-ated during periods of low demand to pump water from a lower reservoir to an upper reservoir. PHES remains an important technology for ES and is expected to play a noteworthy part in the transition to a more sustainable energy upcoming" Table 1 should be moved to this section.  - Text from "ESTs play a critical role in achieving a sustainable energy future. They enable energy to be stored and dispatched when it is needed, improving the reliability, resilience, and efficiency of energy systems. MCDM methods have been developed to help assess and compare the different ESTs." the end should be part of the new section Literature review. Also, should be described more similar studies newer dates.  - Section Methods. Subsection 2.1. and 2.2. should be replaced. The first should show steps of the Entropy method, then VIKOR according to the order of applying these methods in the MCDM model. - Ranking of Energy Storage Technologies. Before you show Table 2 it is necessary to give a description above Table. Also, in Table 3. - It isn't clear how you select criteria. Please provide a description. - "Decision Matrix (X) is created by considering data available from Table 1 and using the Delphi method [42]. The final data is agreed one by five academic experts as shown in Table 4." This should be explained in more detail. Also, should be shown data about five academic experts. - It needs to add new papers in References to capture hot novelties. - The rest of the manuscript is well structured, sensitivity analysis provided and results validated.

Author Response

Sr. No.

Comments

Corrective action and Response

1

- Language in the manuscript in a few places needs to be improved.

We checked the language with Grammarly and corrected it wherever necessary.

2

- The abstract is missing some core elements. Please add novelty, contribution, and results in the short term.

The abstract has been rewritten to add novelty, contribution, and results.

3

- Overall structure of the paper isn’t appropriate. The introduction is too long and contains one Figure and one Table. It must be restructured in the following way:

In the introduction, you should add clear goals of the research, contributions, and reasons for writing this study.

The second section should be a Literature review (Background), and many current introductions can be moved into this section.

The third section can be Materials and methods.

In this part of the manuscript should be moved Figure 1, and Table 1. Also, the text beginning from “Pumped Hydro Energy Storage (PHES): PHES works by using excess electricity generated during periods of low demand to pump water from a lower reservoir to an upper reservoir. PHES remains an important technology for ES and is expected to play a noteworthy part in the transition to a more sustainable energy upcoming” Table 1 should be moved to this section. 

 - Text from “ESTs play a critical role in achieving a sustainable energy future. They enable energy to be stored and dispatched when it is needed, improving the reliability, resilience, and efficiency of energy systems. MCDM methods have been developed to help assess and compare the different ESTs.” the end should be part of the new section Literature review. Also, should be described more similar studies newer dates. 

The overall structure of the paper is revised now as per your suggestions. Thanks for nice recommendations.

4

- Section Methods. Subsection 2.1. and 2.2. should be replaced. The first should show steps of the Entropy method, then VIKOR according to the order of applying these methods in the MCDM model.

Materials and Methods Section is now revised as per suggestions.

5

- Ranking of Energy Storage Technologies. Before you show Table 2 it is necessary to give a description above Table. Also, in Table 3.

Descriptions of the energy storage technologies and attributes are added as suggested.

6

- It isn’t clear how you select criteria. Please provide a description.

Description of criteria is now added.

7

- “Decision Matrix (X) is created by considering data available from Table 1 and using the Delphi method [42]. The final data is agreed one by five academic experts as shown in Table 4.” This should be explained in more detail. Also, should be shown data about five academic experts.

Delphi's method is explained in detail.

8

- It needs to add new papers in References to capture hot novelties.

Thanks for suggestions, Literature review is now enriched with latest references.

9

- The rest of the manuscript is well structured, sensitivity analysis provided and results validated.

Thanks for nice comments.

Reviewer 3 Report

Thank you for inviting me as a reviewer for the paper titled Sustainability Assessment of Energy Storage Technologies based on Commercialization Viability: MCDM Model.

This research performs the performance evaluation (ranking) of various energy storage technologies. For ranking, the authors use the MCDM model based on the Entropy method (for defining weighting coefficients of criteria) and VIKOR method (for ranking alternatives - energy storage technologies). The problem that is solved in the paper is exciting and current. The methodology applied in this paper is well-known in the literature. The paper has the potential to be published.

The authors need to consider the following major points as a limitation or further scope for refining the paper:

1) Rewrite the abstract - In the abstract, the authors should describe the model in more detail, i.e., which methods they used in the MCDM model. It should also emphasize that a sensitivity analysis has been performed. In the end, one sentence should highlight the result of the research.

2) Add keywords: Entropy and VIKOR.

3) The introduction is too long. The part related to the description of the alternatives (energy storage technologies) should not be in the introduction. It should be written about in a separate section (for example, section: Description of the problem).

4) The paper's other sections should be discussed in one paragraph at the end of the introduction.

5) Separate the introductory part and the literature analysis (LA) section. The analysis of the literature is a very weak point of this paper and must be improved. Add another papers (10-15) of more recent date (period 2021-2023), such as: Puška, A., Štilić, A., & Stojanović, I. (2023). Approach for multi-criteria ranking of Balkan countries based on the index of economic freedom. Journal of Decision Analytics and Intelligent Computing, 3(1), 1–14. https://doi.org/10.31181/jdaic10017022023p; Yildirim, B. F., & Kuzu Yıldırım, S. (2022). Evaluating the satisfaction level of citizens in municipality services by using picture fuzzy VIKOR method: 2014-2019 period analysis. Decision Making: Applications in Management and Engineering, 5(1), 50-66. https://doi.org/10.31181/dmame181221001y. Based on literature analysis authors should better highlight the objective of their paper and to what extent it contributes to close a gap in the existing literature and/or practice. What is the innovative value of the contribution proposed by the authors? This is an essential part of the LA section.

6) Show a flowchart with all phases and steps of the model in one figure (in section Methods).

7) Check the text from line 264 to line 266. Check the equatation in line 264 (Q (Rank2) − Q (Rank1) ≥ (1/(n −1)). The original VIKOR method defines that the difference should be ≥min(0.25; 1/(n −1)). Also, with the original method, this difference is also calculated for the other alternatives, not only as the difference between the first and the second.

8) Explain each criterion in one to two sentences. How the authors defined the criteria (by analyzing the literature, using experts, etc.).

9) Line 315 - the authors use the term "scheme" instead of "step."

10) In conclusion, something should be said about the model.

11) Show the limitations of the proposed methodology and this study.

12) Add future research.

Please, mark the requested changes, in the corrected version, in a different color.

Author Response

Sr. No.

Comments

Corrective Action and Response

1

1) Rewrite the abstract - In the abstract, the authors should describe the model in more detail, i.e., which methods they used in the MCDM model. It should also emphasize that a sensitivity analysis has been performed. In the end, one sentence should highlight the result of the research.

The abstract has been rewritten. The sensitivity analysis and result part has been added.

2

2) Add keywords: Entropy and VIKOR.

Added.

3

3) The introduction is too long. The part related to the description of the alternatives (energy storage technologies) should not be in the introduction. It should be written about in a separate section (for example, section: Description of the problem).

Introduction shortened and separate sections created.

4

4) The paper’s other sections should be discussed in one paragraph at the end of the introduction.

Added

5

5) Separate the introductory part and the literature analysis (LA) section. The analysis of the literature is a very weak point of this paper and must be improved. Add another papers (10-15) of more recent date (period 2021-2023), such as:

Puška, A., Štilić, A., & Stojanović, I. (2023). Approach for multi-criteria ranking of Balkan countries based on the index of economic freedom. Journal of Decision Analytics and Intelligent Computing, 3(1), 1–14. https://doi.org/10.31181/jdaic10017022023p;

Yildirim, B. F., & Kuzu Yıldırım, S. (2022). Evaluating the satisfaction level of citizens in municipality services by using picture fuzzy VIKOR method: 2014-2019 period analysis. Decision Making: Applications in Management and Engineering, 5(1), 50-66. https://doi.org/10.31181/dmame181221001y.

 Based on literature analysis authors should better highlight the objective of their paper and to what extent it contributes to close a gap in the existing literature and/or practice. What is the innovative value of the contribution proposed by the authors? This is an essential part of the LA section.

Introduction part has been separated from the rest of the section.

Few current research papers have been added.

The suggested references are useful and cited at suitable places.

6

6) Show a flowchart with all phases and steps of the model in one figure (in section Methods).

Thanks for wonderful suggestion, now flowchart with all phases and steps of the model is incorporated in the paper.

7

7) Check the text from line 264 to line 266. Check the equatation in line 264 (Q (Rank2) − Q (Rank1) ≥ (1/(n −1)). The original VIKOR method defines that the difference should be ≥min(0.25; 1/(n −1)). Also, with the original method, this difference is also calculated for the other alternatives, not only as the difference between the first and the second.

The original VIKOR did contain steps, as said. However, in some of the current research papers and books referred to, the condition of (Q (Rank2) − Q (Rank1) ≥ (1/(n −1)) is only mentioned as a sufficient condition.

8

8) Explain each criterion in one to two sentences. How the authors defined the criteria (by analyzing the literature, using experts, etc.).

Criteria have been identified through the literature review. Each criterion has been discussed in the section on Energy Storage Technologies.

9

9) Line 315 - the authors use the term “scheme” instead of “step.”

Scheme changed to Step.

10

10) In conclusion, something should be said about the model.

Added

11

11) Show the limitations of the proposed methodology and this study.

Added

12

12) Add future research.

Please, mark the requested changes, in the corrected version, in a different color.

Future scope added.

All changes are highlighted with red text.

Round 2

Reviewer 2 Report

Can be published.